# The epidemiology and evolution of IgA nephropathy over two decades: A single centre experience

Joshua Storrar [1,2]*, Rajkumar Chinnadurai[2], Smeeta Sinha[2], Philip A. Kalra[2]

**1** Faculty of Biology, Medicine and Health, University of Manchester, Manchester, United Kingdom, **2** Renal Department, Northern Care Alliance NHS Foundation Trust, Salford Royal Hospital, Salford, United Kingdom

* joshua.storrar2@nca.nhs.uk

## Abstract

### Background and objectives

IgA nephropathy (IgAN) is the most common glomerulonephritis worldwide, with an incidence of 2.5 per 100,000 population per year. The 10-year risk of progression to end stage kidney disease (ESKD) or halving of eGFR is 26%. Here we aimed to collect a comprehensive dataset of IgAN patients at our centre over 2 decades to provide real world data, describe outcomes and determine the effects of immunosuppression use.

### Design, setting, participants and measurements

All patients diagnosed with biopsy-proven IgAN at our centre over 2 decades were identified. After exclusions, the total cohort size was 401. Data relating to (i) baseline demographics, (ii) laboratory and urine results, (iii) histological data, and (iv) outcomes of initiation of renal replacement therapy (RRT) and mortality were collected.

### Results

The median age was 45.0 years, with 69.6% male and 57.6% hypertensive; 20.4% received immunosuppression, 29.7% progressed to RRT and 19.7% died, over a median follow up period of 51 months. Baseline eGFR was 46.7ml/min/1.73m$^2$ and baseline uPCR was 183mg/mmol. Median rate of eGFR decline was -1.31ml/min/1.73m$^2$/year. Those with a higher MEST-C score had worse outcomes. Immunosuppression use was associated with an increased rate of improvement in proteinuria, but not with a reduction in RRT or mortality. Factors favouring improved outcomes with immunosuppression use included female gender; lower age, blood pressure and T-score; higher eGFR; and ACEi/ARB use.

### Conclusions

A variety of clinical and histological factors are important in determining risk of progression in IgAN. Therapeutic interventions, particularly use of immunosuppression, should be individualised and guided by these factors.

**Data Availability Statement:** All relevant data are within the paper and its Supporting Information files.

**Funding:** The Salford glomerulonephritis research group was generously supported by an

unrestricted project grant from Vifor. https://www.viforpharma.com/ The funders had no role in study design, data collection and analysis, decision to publish, or preparation of the manuscript.

**Competing interests:** I have read the journal's policy and the authors of this manuscript have the following competing interests. S Sinha has received grants from Johnson and Johnson and AstraZeneca; speaker and lecture fees from AstraZeneca, Napp, Bayer, Sanofi-Genzyme, Novartis, and Vifor Pharma; is on advisory boards for Novartis, AstraZeneca, Bayer, and Travere; and has a clinical consultancy role with Sanifit. PA Kalra has received grants from Astellas, Vifor Pharma, BergenBio and Evotec; speaker and lecture fees from AstraZeneca, Napp, Bayer, Novartis, Vifor Pharma, Pharmacosmos, Boehringer Ingelheim; is on advisory boards for AstraZeneca and Vifor Pharma; and has a consultancy role with Bayer, Astella, Otsuka and Unicyte. JS and RC have no conflicts of interest to report. These do not alter out adherence to PLOS ONE policies on sharing data and materials

# Introduction

IgA Nephropathy (IgAN) is the most common glomerulonephritis worldwide [1]. It was first described in 1968 by Jean Berger, a French renal histopathologist who initially gave his name to the disease, and Nicole Hinglais [2]. IgAN has an incidence of at least 2.5 per 100,000 population per year [1]. There is significant geographical variation with an increased prevalence in Far East Asia compared to Europe, whilst in Africa it is even less prevalent. Male: female ratio is 3:1 in Europeans but 1:1 in East Asians [3]. Presentation ranges from isolated haematuria to significant proteinuria to acute kidney injury (AKI) and even chronic kidney disease (CKD). The 10-year risk of progression to end stage kidney disease (ESKD) or halving of GFR is 26% [4]. The difficulty lies in predicting the rate of renal decline on an individual basis, and as such determining what treatment to use.

In the pathogenesis of IgAN, the normal physiological process of IgA production becomes dysregulated, through a mechanism that remains unclear. A four-hit hypothesis is widely accepted [5].

Until the late 2000s, there was no agreed consensus on how to consider histology findings when predicting an individual's risk of progression to ESKD. In 2009, this changed when the Oxford classification of IgAN was published [6]. It identified 4 variables that had independent value in predicting renal outcome: mesangial hypercellularity (M), endocapillary hypercellularity (E), segmental glomerulosclerosis (S), and tubular atrophy and interstitial fibrosis (T). Subsequently, in 2017, a C score was added to indicate the presence of crescents [7], hence the MEST-C score was developed.

There are many variables that determine an individual's risk of progression in IgAN. These include age, sex, ethnicity, proteinuria, eGFR, blood pressure, MEST-C score and use of immunosuppression and renin-angiotensin system (RAS) blockade at or prior to biopsy. In an effort to combine these variables in a clinically meaningful way a risk prediction tool was created, known as the International Risk-Prediction Tool in IgA Nephropathy [8]. It calculates the risk of a 50% decline in eGFR or progression to ESKD up to 7 years post biopsy. It should be noted that it is only the MEST score (with no information regarding the presence or absence of crescents) and not the MEST-C which is included in this tool, due to the fact that the C score added no value in addition to the inclusion of race(8).

Since the first description of IgAN more than 50 years ago, the mainstay of management has been conservative treatment, primarily by optimising blood pressure and proteinuria with RAS blockade. Given that the pathogenesis involves the formation of immune complexes, there has been a long history of using steroids and other immunosuppressants to treat IgAN. However, to date there has been no convincing published evidence to suggest that any immunosuppressant is effective, when weighed up against their potential adverse effects (see the VALIGA, STOP-IgAN and TESTING landmark clinical trials [4, 9, 10]).

Here, we collected a large clinical dataset of all patients with IgAN at our centre over a 20-year period, with the aim to describe the epidemiology of our cohort, to determine variation in management and to compare outcomes.

# Materials and methods

## Sampling

This was a retrospective observational longitudinal study conducted on patients diagnosed with IgAN at our tertiary renal centre (Salford Royal Hospital, UK), encompassing a catchment population of 1.55 million, between January 2000 and December 2019. The population is largely urban, with a mixture of affluent areas and those with increased social deprivation

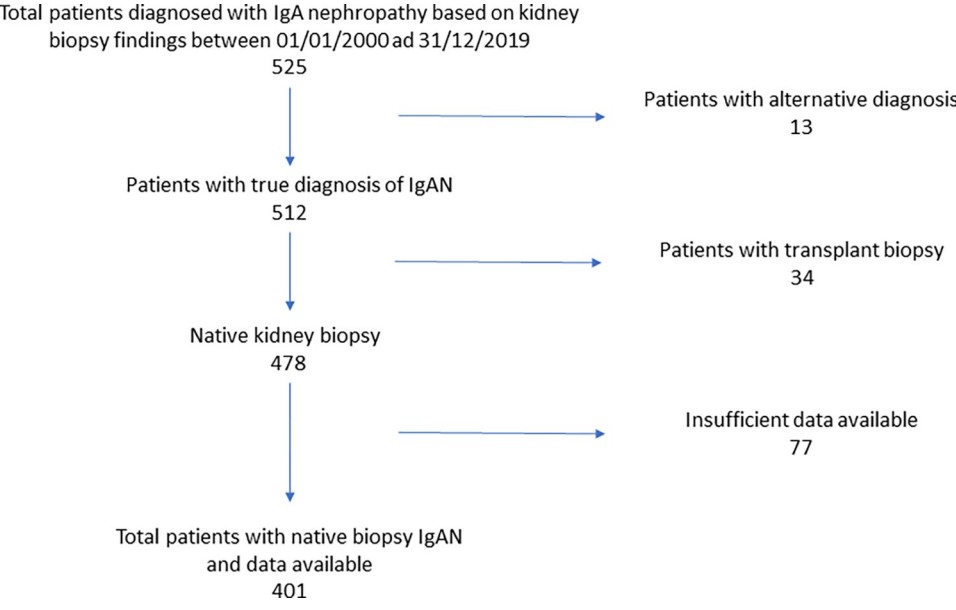

**Fig 1. Patient recruitment to the study.**

The Salford Royal Kidney Biopsy database was screened for patients with a diagnosis of IgAN between January 2000 and December 2019. This initially showed a total of 525 patients. Patients were excluded if their diagnosis was not actually IgAN (13), had a transplant biopsy rather than a native biopsy (34), or did not have sufficient clinical data available (77). The final study population was 401 patients (Fig 1).

MEST-C score was routinely recorded as part of the clinical biopsy assessment from 2012 onwards. For those patients whose biopsy was undertaken prior to 2012, the biopsy report was reviewed by one of the authors (JS) and a MEST-C score was determined; 10% of these scores were subsequently validated by an independent nephrologist. Comparative analysis was made between patients in five groups with different MEST-C scores.

The date of kidney biopsy was used as the study baseline, and all patients were followed until they reached a study endpoint which included (i) commencement of RRT, (ii) death, (iii) end of analysis period (31 December 2020) or (iv) lost to follow up or last documented clinic appointment.

Data on baseline characteristics, laboratory results, treatment received to include RAS blockade and immunosuppression (started at any point in the patient journey), date of initiation of RRT (either transplantation or dialysis) and mortality were gathered from the electronic patient record (EPR). All baseline characteristics and laboratory results were those obtained at the time of biopsy or within 6 months. Serial values were obtained for eGFR and uPCR to allow calculation of the change in these parameters over time.

Immunosuppression treatment included prednisolone, cyclophosphamide, tacrolimus, ciclosporin, azathioprine and MMF.

A comorbidity of hypertension was defined as a history of hypertension recorded in hospital records, and/or being on antihypertensive therapy. A comorbidity of cardiovascular disease included a history of ischaemic heart disease, heart failure, cerebrovascular disease, or peripheral vascular disease.

Estimated glomerular filtration rate (eGFR) values were calculated by the CKD Epidemiology Collaboration (CKD-EPI) formula.

## Ethical considerations

The study complies with the declaration of Helsinki and as indicated by the NHS Health Research Authority online tool http://www.hra-decisiontools.org.uk/research this study was not considered research requiring research ethics committee review as it was a retrospective observational study using measurements routinely collected and using fully anonymised and secondary use of data. The need for individual patient consent was waived by the Research and Innovation committee of the Northern Care Alliance NHS Group.

The committee granted study approval and registered the study (Ref: ID S21HIP40) after approving the methodological protocol as outlined above.

## Statistical analysis

Analysis of baseline characteristics, comorbidities, MEST-C score, requirement for RRT, mortality, use of RAS blockade and use and effect of immunosuppression was undertaken in the total cohort. Continuous non-parametric variables are presented as median (interquartile range) and the Mann-Whitney U-test was used to test statistical significance. Categorical data are expressed as percentage, and the Chi-square test was used to test statistical significance.

The association of baseline variables with requirement for RRT and mortality was calculated using univariate and multivariate Cox proportional hazard models to determine hazard ratios (HRs), 95% confidence intervals (CIs) and statistical significance.

CKD progression in the overall cohort was computed using the rate of change of eGFR (delta eGFR) from baseline to study end point, with the linear regression slope generated using all available eGFR measurements (using a minimum of 3 eGFR values and a minimum follow up duration of 12 months). Similarly, the rate of change of uPCR (delta uPCR) from baseline to study endpoint was calculated using linear regression from serial uPCR measurements. The Mann-Whitney U-test was used to compare statistical significance between the groups.

The effect of immunosuppression was determined by comparing those who received immunosuppression and those who did not. Further analysis was performed by propensity score matching those patients receiving immunosuppression 1:1 with non-immunosuppressed patients matched for hypertension, baseline creatinine and proteinuria based on a priori from previous observations. As the cohorts were already well matched for age and gender these variables were not included. Propensity scores were generated using binary logistic regression analysis using a nearest neighbour approach.

A p value <0.05 was considered statistically significant throughout the analysis. All statistical analysis was performed using IBM SPSS (version 24, University of Manchester).

## Results

### Characteristics of the overall cohort

A total of 401 patients had available data for analysis in the study. Table 1 depicts the baseline characteristics of the full cohort in the first column. The median age was 45.0 years (30–61), 69.6% were male and 87.5% were Caucasian. 7.5% were diabetic, 57.6% hypertensive and 9.2% had co-existing cardiovascular disease. Baseline blood results showed a median creatinine of 142μmol/L (91–241), median eGFR 46.7ml/min/1.73m$^2$ and median uPCR of 183mg/mmol (76–401). The median rate of decline of eGFR was -1.31ml/min/1.73m$^2$/year (-5.6 to 0.67) and the median change in uPCR was -4.46mg/mmol/year (-22.7 to 5.5). RAS blockade was used in 79.6% and immunosuppression in 20.4%. Progression to ESKD requiring RRT was seen in 29.7% of our cohort, and the mortality rate was 19.7%. The median follow-up duration was 51

**Table 1. Comparison of baseline characteristics and outcomes based on MEST-C score category.**

| | Total n = 401 | MEST-C score 0 (n = 62) | MEST-C score 1 (n = 102) | MEST-C score 2 (n = 107) | MEST-C score 3 (n = 83) | MEST-C score >3 (n = 47) | P-value |
|---|---|---|---|---|---|---|---|
| Age, years | 45 (30–61) | 44 (28–58.3) | 50.0 (29.0–66.0) | 44 (31–58) | 41.0 (29.0–54.0) | 47.0 (33.0–66.0) | 0.122 |
| Male | 279 (69.6) | 44 (71.0) | 65 (63.7) | 76 (71.0) | 57 (68.7) | 37 (78.7) | 0.448 |
| Caucasians | 351 (87.5) | 51 (82.3) | 94 (92.2) | 93 (86.9) | 74 (89.2) | 39 (83.0) | 0.538 |
| Diabetes | 30 (7.5) | 3 (4.8) | 9 (8.8) | 8 (7.5) | 9 (10.8) | 1 (2.1) | 0.381 |
| Hypertension | 231 (57.6) | 24 (38.7) | 60 (58.8) | 56 (52.3) | 52 (62.7) | 39 (83.0) | **<0.001** |
| CVD | 37 (9.2) | 4 (6.5) | 8 (7.8) | 10 (9.3) | 10 (12.0) | 5 (10.6) | 0.790 |
| SBP, mmHg | 132 (122–143) | 130 (116.8–140) | 131.5 (125–144.25) | 130 (120–141) | 132 (122–143) | 139.5 (128.8–145) | **0.002** |
| DBP mmHg | 80 (70–87) | 79.5 (70–85) | 80.0 (70–87.3) | 79 (70–85) | 80 (70–90) | 82 (76.8–90) | **0.022** |
| Creatinine, μmol/L | 142 (91–241) | 90.5 (74.25–118.9) | 119 (79–191.5) | 140 (93.5–191.5) | 218.5 (133–314.5) | 224 (158–370) | **<0.001** |
| eGFR, ml/min/1.73m$^2$ | 46.7 (24.7–82.2) | 85.6 (53.2–106.1) | 57.5 (27.6–90.3) | 48.5 (31.7–74.8) | 29.4 (17.1–55.2) | 27.8 (15.9–40.4) | **<0.001** |
| uPCR, mg/mmol | 183 (76–401) | 53 (18.5–241.5) | 117 (57–285) | 167 (86.75–333) | 260 (167–522) | 321.5 (207.3–635) | **<0.001** |
| IgA, g/L | 3.92 (2.96–5.14) | 3.99 (2.94–4.65) | 4.11 (3.17–5.64) | 4.46 (3.28–5.76) | 3.62 (2.84–4.77) | 3.30 (2.63–4.55) | 0.320 |
| C3, g/L | 1.21 (1.00–1.42) | 1.31 (1.06–1.46) | 1.30 (1.07–1.53) | 1.18 (1.00–1.38) | 1.09 (0.93–1.39) | 1.15 (1.01–1.32) | 0.061 |
| Haemoglobin, g/L | 124 (108–141) | 138 (119.8–153.3) | 125 (106–140.8) | 129 (119–144) | 116 (101–132) | 113.5 (99.5–128) | **<0.001** |
| Albumin, g/L | 39 (34–43) | 41.5 (37.8–44) | 40 (34–43) | 40 (34–43) | 38 (34–42) | 37 (31.5–42) | **0.010** |
| ALP, U/L | 71 (60–90) | 65 (57.5–86.5) | 73 (59–94.5) | 69 (60–82) | 73 (62–96) | 75.5 (60–100.5) | 0.107 |
| P04, mmol/L | 1.21 (1.03–1.41) | 1.14 (1.00–1.26) | 1.15 (1.03–1.41) | 1.15 (0.99–1.28) | 1.30 (1.08–1.57) | 1.50 (1.18–1.76) | **<0.001** |
| CCa, mmol/L | 2.27 (2.13–2.33) | 2.29 (2.26–2.38) | 2.27 (2.16–2.33) | 2.28 (2.20–2.37) | 2.19 (2.05–2.29) | 2.13 (2.02–2.29) | **0.001** |
| Delta eGFR, ml/min/1.73m$^2$/year | -1.31 (-5.6–0.67) | 0.38 (-2.37–2.44) | -1.21 (-5.05–1.20) | -1.22 (-4.25–0.08) | -2.16 (-7.94–0.47) | -3.57 (-9.34- -1.18) | **<0.001** |
| Delta uPCR, mg/mmol/year | -4.46 (-22.7 to 5.5) | -1.26 (-6.98–0.95) | -4.86 (-27.9–11.5) | -2.08 (-15.63–11.8) | -10.8 (-44.1- -0.26) | -10.1 (-34.9–5.66) | 0.119 |
| ACEi/ ARB | 319 (79.6) | 43 (70.5) | 82 (80.4) | 92 (86.8) | 68 (81.9) | 34 (75.6) | 0.120 |
| Immunosuppression | 82 (20.4) | 4 (4.9) | 20 (19.6) | 20 (18.7) | 24 (28.9) | 14 (29.8) | **0.008** |
| RRT | 119 (29.7) | 2 (3.2) | 16 (15.7) | 26 (24.3) | 40 (48.2) | 29 (61.7) | **<0.001** |
| Mortality | 79 (19.7) | 8 (12.9) | 27 (26.5) | 18 (16.8) | 12 (14.5) | 14 (29.8) | **0.044** |
| Follow up duration, months | 51 (18–97.5) | 49 (26.8–99.5) | 43 (19–89.5) | 82 (31–121) | 52 (16–79) | 21 (4–54) | **0.004** |

Continuous variables are presented as median (interquartile range), p-value by Mann–Whitney U-test. Categorical variables presented as number (percentage), p-value by Chi-squared test.

p-value comparing the groups MEST-C score> 3 and MEST-C score 0.

ACEi, angiotensin converting enzyme inhibitor; ALP, alkaline phosphatase; ARB, angiotensin receptor blockade; C3, complement 3; CCa, corrected calcium; CVD, cardiovascular disease; DBP, diastolic blood pressure; eGFR, estimated glomerular filtration rate; IgA, immunoglobulin; P04, phosphate; uPCR, urine protein creatinine ratio; RRT, renal replacement therapy; SBP, systolic blood pressure.

months (18–97.5), with the end point of the study being either last recorded follow up, date of initiation of RRT or death.

## Analysis according to total MEST-C score

The overall cohort of 401 patients was divided into five groups according to their MEST-C score: score 0 (n = 62), score 1 (n = 102), score 2 (n = 107), score 3 (n = 83) and score >3 (n = 47). The baseline characteristics for these groups can be seen in Table 1. As the MEST-C score increased, there were increased rates of hypertension; higher creatinine, uPCR, and

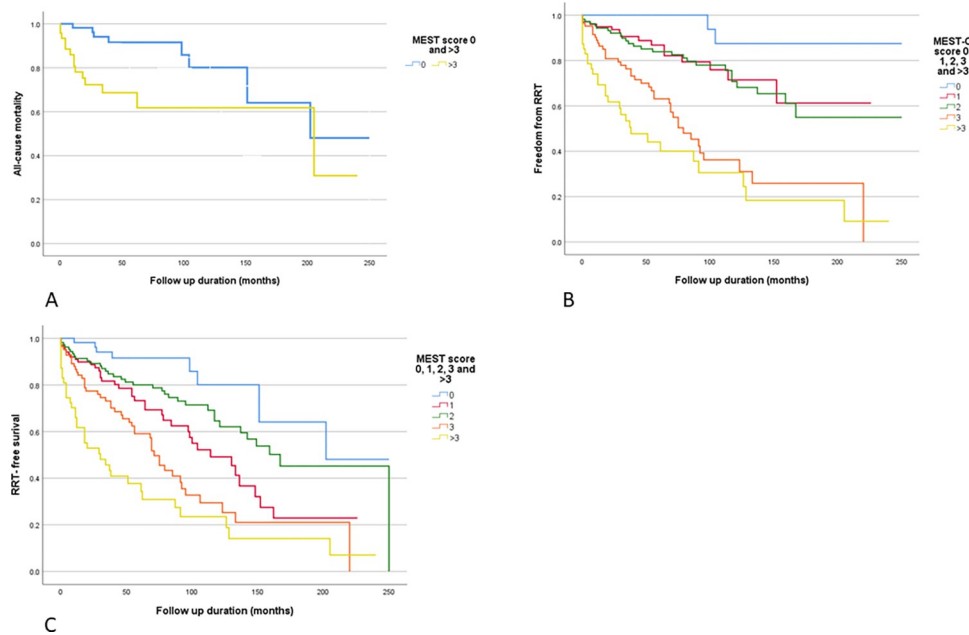

**Fig 2.** Kaplan-Meier curves for all-cause mortality (A), freedom from RRT (B) and RRT-free survival (C) for different MEST-C score groups. P-values 0.002, <0.001 and <0.001 respectively.

phosphate values; increased immunosuppression use and requirement for RRT; along with lower haemoglobin, eGFR, albumin and calcium values. The highest score group was then compared to the lowest score group. There was significantly more hypertension observed in the >3 score group (83%) compared to the 0-score group (38.7%). A variety of markers associated with a more advanced stage of kidney disease were also observed in the highest score group (lower haemoglobin, albumin and calcium, higher phosphate, creatinine, and urine PCR). There was a greater reduction in uPCR over time in the >3-score group compared to the 0-score group (-10.1 vs –1.26mg/mmol/year) although this was not significant (p = 0.119). There was also a greater degree of renal function decline in the >3-score group than in the 0-score group (-3.57 vs 0.38ml/min/1.73m$^2$/year, p<0.001). Immunosuppression was used sparingly in the MEST-C 0-score group (4.9%), with increasing rates in the higher scoring groups (19.6%, 18.7%, 28.9% and 28.9% for MEST-C groups 1, 2, 3 and > 3, respectively). Need for RRT increased across all 5 groups (3.2%, 15.7%, 24.3%, 48.2% and 61.7% respectively —p<0.001 when first group compared to the last), whilst mortality was more variable (12.9%, 26.5%, 16.8%, 14.5% and 29.8% for MEST-C groups 0, 1, 2, 3, > 3, respectively). There was a difference in mortality rates between the highest and lowest scoring group, p = 0.044. This is also demonstrated in Fig 2 which shows Kaplan-Meier Curves for all-cause mortality (A), freedom from RRT (B) and RRT-free survival (C) split according to the 5 MEST-C score groups. Whilst in Fig 2A, representing all-cause mortality, there was convergence, this occurred late in the follow up duration, and a statistically significant difference between the groups was maintained.

## All-cause mortality and need for RRT

Cox regression analysis was performed to determine factors at baseline which were associated with mortality and need for RRT (Table 2). A univariate model showed the following factors to be significantly associated with mortality: increasing age, non-Caucasian ethnicity, diabetes,

**Table 2. Association between baseline variables and all-cause mortality and need for RRT utilising univariate and multivariate cox regression.**

| | All-cause mortality | | | | Need for RRT | | | |
| --- | --- | --- | --- | --- | --- | --- | --- | --- |
| | Univariate model | | Multivariate model | | Univariate model | | Multivariate model | |
| | Hazard ratio (95% CI) | P-value | Hazard ratio (95% CI) | P-value | Hazard ratio (95% CI) | P-value | Hazard ratio (95% CI) | P-value |
| Age, years | 1.08 (1.06–1.09) | <**0.001** | 1.03 (1.01–1.06) | **0.004** | 1.00 (0.99–1.01) | 0.993 | - | - |
| Male | 1.55 (0.91–2.62) | 0.102 | - | - | 1.19 (0.79–1.81) | 0.390 | - | - |
| Caucasian | 0.24 (0.06–0.97) | **0.045** | 0.31 (0.04–2.33) | 0.257 | 1.07 (0.61–1.87) | 0.820 | - | - |
| Diabetes | 1.66 (1.24–2.23) | **0.001** | 1.81 (1.26–2.60) | **0.001** | 0.95 (0.64–1.39) | 0.775 | - | - |
| Hypertension | 2.02 (1.23–3.28) | **0.005** | 1.56 (0.75–3.24) | 0.233 | 2.88 (1.85–4.49) | <**0.001** | 1.82 (0.98–3.35) | 0.058 |
| CVD | 4.3 (2.62–7.05) | <**0.001** | 1.78 (0.92–3.41) | 0.086 | 0.98 (0.49–1.95) | 0.966 | - | - |
| SBP at biopsy, mmHg | 1.03 (1.02–1.04) | <**0.001** | 1.01 (0.99–1.02) | 0.299 | 1.02 (1.01–1.03) | **0.004** | 1.00(0.98–1.01) | 0.528 |
| DBP at biopsy, mmHg | 1.00 (0.97–1.02) | 0.995 | - | - | 1.03 (1.01–1.05) | **0.001** | 1.03 (1.00–1.05) | **0.041** |
| M1 | 0.91 (0.58–1.41) | 0.667 | - | - | 1.184(0.82–1.72) | 0.375 | - | - |
| E1 | 2.35 (1.16–4.74) | **0.017** | 1.63(0.59–4.48) | 0.348 | 1.52 (0.77–3.02) | 0.231 | - | - |
| S1 | 0.77 (0.49–1.20) | 0.254 | - | - | 1.53 (1.04–2.262) | **0.031** | 0.88 (0.49–1.56) | 0.659 |
| T1/2 | 1.40 (1.06–1.86) | **0.018** | 0.83(0.59–1.12) | 0.277 | 2.84 (2.264–3.56) | <**0.001** | 1.60 (1.01–2.55) | **0.045** |
| C1 | 1.24 (0.82–1.89) | 0.304 | - | - | 1.55 (1.13–2.11) | **0.006** | 1.21 (0.70–2.10) | 0.492 |
| Total MEST score | 1.15 (0.97–1.36) | 0.117 | - | - | 1.77 (1.54–2.03) | <**0.001** | 1.16 (0.81–1.65) | 0.418 |
| eGFR at biopsy, ml/min/1.73m2 | 0.96 (0.95–0.97) | <**0.001** | 0.97 (0.95–0.99) | **0.003** | 0.97 (0.96–0.97) | <**0.001** | 0.99 (0.98–1.01) | 0.194 |
| Creatinine at biopsy, μmol/L | 1.01 (1.01–1.01) | <**0.001** | 1.00 (0.99–1.00) | 0.726 | 1.00 (1.002–1.003) | <**0.001** | 1.001 (1.000–1.003) | 0.104 |
| uPCR at biopsy, mg/mmol | 1.01 (1.01–1.02) | <**0.001** | 1.00 (1.00–1.00) | 0.241 | 1.00 (1.001–1.002) | <**0.001** | 1.001 (1.000–1.002) | **0.002** |
| ACEi/ ARB use | 0.27 (0.17–0.43) | <**0.001** | 0.55 (0.28–1.08) | 0.082 | 0.456 (0.30–0.70) | <**0.001** | 0.48 (0.28–0.84) | **0.009** |
| Immunosuppression use | 1.31 (0.76–2.25) | 0.328 | - | - | 1.281 (0.82–2.00) | 0.276 | - | - |

Multivariate model for all-cause mortality adjusted for age, ethnicity, diabetes, hypertension, CVD, SBP at biopsy, E score, T score, creatine at biopsy, uPCR at biopsy and ACEi/ARB use. Multivariate model for need for RRT adjusted for hypertension, SBP at biopsy, DBP at biopsy, S, T, C score, total MEST-C score, creatinine at biopsy, uPCR at biopsy and ACEi/ARB use.

ACEi, angiotensin converting enzyme inhibitor; ARB, angiotensin receptor blocker; C, crescent; CVD, cardiovascular disease; DBP, diastolic blood pressure; E, endocapillary hypercellularity; eGFR, estimated glomerular filtration rate; M, mesangial hypercellularity; S, segmental sclerosis; SBP, systolic blood pressure; T, tubular atrophy and interstitial fibrosis; uPCR, urine protein creatinine ratio.

hypertension, cardiovascular disease, systolic blood pressure, endocapillary hypercellularity (E) score, tubular atrophy and interstitial fibrosis (T) score (T1/2), creatinine, uPCR, and ACEi/ARB use. Multiple factors were associated with need for RRT: hypertension, systolic and diastolic blood pressure, S, T and C score, total MEST-C score, creatinine, uPCR and ACEi/ARB use. Immunosuppression use was not found to be a factor associated with all-cause mortality or need for RRT.

Multivariate cox regression determined those factors with a positive correlation in the univariate model which remained significant (Table 2). For all-cause mortality these factors were age (HR 1.05, p<0.001), diabetes (HR 1.70, p = 0.003) and creatinine (HR 1.00, p = 0.01). With need for RRT, these factors were hypertension (HR 2.14, p = 0.011), diastolic blood pressure (HR 1.03, p = 0.05), T1/2 (HR 1.76, p = 0.01), creatinine (HR 1.002, p<0.001), uPCR (HR 1.001, p<0.001) and ACEi/ARB use (HR 0.48, p = 0.01).

## Effect of immunosuppression

A variety of different types of immunosuppression were recorded including IV cyclophosphamide and prednisolone (n = 24), prednisolone alone (n = 30), prednisolone and MMF (n = 12)

and several combinations of agents. For this analysis only those patients who had a delta eGFR result available (n = 346) were included. We initially compared those who received immunosuppression (n = 69) versus all of those who did not (n = 277); Table 3. Those given immunosuppression were more likely to have an E1 score (17.4% vs 4.0%, p<0.001), C1 score (37.7% vs 10.8%, p<0.001) and total MEST-C score of >2 (42% vs 29.2%, p = 0.041). They also had a lower IgA level and calcium level, and a higher uPCR. They showed a greater reduction in proteinuria over time (delta uPCR -16.8 vs -2.65 mg/mmol/year), but there was no difference in eGFR decline (-1.18 vs -1.32ml/min/1.73m$^2$, p = 0.703). There was also no difference in need for RRT or mortality between the two groups.

In the propensity score matched analysis there were 57 patients who received immunosuppression compared with 57 patients matched for hypertension, baseline creatinine and uPCR who did not receive immunosuppression, Table 3. Whilst there was a higher percentage who required RRT (45.6% vs 31.6%) and a higher mortality rate (22.8% vs 12.3%) in the non-immunosuppression group, this did not reach statistical significance. It is also worth noting that the follow up duration was not equal (64 months for immunosuppressed vs 39 months for non-immunosuppressed). Fig 3 depicts Kaplan-Meier curves for all-cause mortality (A), freedom from RRT (B) and RRT-free survival (C), showing separation between the groups but none reaching significance.

We undertook Cox regression analysis to determine those factors at baseline which were associated with a worse outcome in those who received immunosuppression (S1 Table). In the univariate model the following factors were significantly associated with a worse outcome: increasing age, male gender, hypertension, increased systolic blood pressure, T1/2 score, lower eGFR and lack of ACEi/ARB use. In the multivariate model, eGFR (HR 0.98, 95% CI 0.96–0.99, p = 0.004) and ACEi/ARB use (HR 0.40, 95% CI 0.17–0.95, p = 0.037) remained significant.

The supplementary material also includes details of analysis of the group by rate of eGFR decline (S1 File, S2 Table and S1 Fig); outcomes pre and post 2012 (S3 Table); and 5 and 10 year all-cause mortality and freedom from RRT split according to decade and presented as Kaplan-Meier charts (S2 and S3 Figs).

## Discussion

This is one the largest retrospective observational studies assessing clinical and histological characteristics, along with outcomes, for IgAN. This provides important real-world data which will be useful for clinicians, particularly as the IgAN landscape changes with the introduction of novel therapies.

A previous cohort study enrolled 154 patients, and found the average age to be 34 years (younger than our cohort), with 64% male. Jarrick S et al. conducted a Swedish population based study assessing the risk of mortality in over 3622 patients diagnosed with IgAN [11]. They found that over a follow up period of 13.6 years, 577 patients died (15.9%) compared to 2066 (11.5%) in the reference population. This corresponded to a 6 year reduction in life expectancy for those with IgAN. Whilst their rate of 15.9% is not particularly dissimilar to our rate of 19.7%, they do have a longer average duration of follow up (13.6 years). However, the average age of their cohort was lower (median 38.8 years) compared to ours (45.0 years) and so you would expect resultant mortality to be lower. Another study reported an almost double increased risk of death for IgAN patients compared to the general population [12].

A study involving 145 patients in Italy demonstrated that over a mean follow-up of 67 months, 33 (23%) progressed to ESKD, and 61% received some form of immunosuppression. Furthermore, those with a higher time-averaged blood pressure were more likely to progress

**Table 3. Baseline characteristics, laboratory values and outcomes for those who received immunosuppression and those that did not (for all patients with delta eGFR value, n = 346, and for a matched cohort, n = 114).**

| Variable | Unmatched cohort | | | Matched cohort | | |
|---|---|---|---|---|---|---|
| | Immunosuppression (n = 69) | No immunosuppression (n = 277) | P value | Immunosuppression (n = 57) | No immunosuppression (n = 57) | P-value |
| Age, years | 42.0 (31.0–59.5) | 45.0 (29.0–60.0) | 0.988 | 42 (29–57) | 45 (28.5–63) | 0.671 |
| Male | 45 (65.2) | 196 (70.8) | 0.370 | 37 (64.9) | 47 (82.5) | **0.033** |
| Caucasian | 60 (87.0) | 244 (88.1) | 0.835 | 51 (89.5) | 52 (91.2) | 0.494 |
| Diabetes | 22 (2.9) | 22 (7.9) | 0.140 | 2 (3.5) | 5 (8.8) | 0.242 |
| Hypertension | 42 (60.9) | 163 (58.8) | 0.759 | 36 (63.2) | 38 (66.7) | 0.695 |
| CVD | 4 (5.8) | 25 (9.0) | 0.387 | 3 (5.3) | 8 (14) | 0.113 |
| SBP, mmHg | 131.5 (119.25–146.75) | 131 (122.75–142.25) | 0.754 | 131 (119.25–146.75) | 135 (125.5–144.0) | 0.618 |
| DBP, mmHg | 80.0 (70.0–85.0) | 80 (70–88) | 0.569 | 80 (70–85) | 80 (70–88) | 0.909 |
| M 1 | 41 (59.4) | 143 (51.6) | 0.246 | 33 (57.9) | 32 (56.1) | 0.85 |
| E 1 | 12 (17.4) | 11 (4.0) | **<0.001** | 6 (10.5) | 5 (8.8) | 0.751 |
| S 1 | 34 (49.3) | 151 (54.5) | 0.435 | 28 (49.1) | 40 (70.2) | **0.022** |
| T 0 | 48 (69.6) | 160 (57.8) | 0.147 | 38 (66.7) | 22 (19.3) | **<0.001** |
| T 1 | 14 (20.3) | 66 (23.8) | | 14 (24.6) | 15 (26.3) | |
| T 2 | 7 (10.1) | 51 (18.4) | | 5 (8.8) | 20 (35.1) | |
| C 1 | 26 (37.7) | 30 (10.8) | **<0.001** | 21 (36.8) | 11 (19.3) | **0.028** |
| Total MEST-C score (>2) | 29 (42.0) | 81 (29.2) | **0.041** | 20 (35.1) | 31 (54.4) | **0.038** |
| Creatinine at biopsy, μmol/L | 166.5 (94.25–241.75) | 137.0 (90.0–217.5) | 0.096 | 167 (92.5–239.5) | 195 (97–314) | 0.298 |
| uPCR at biopsy, g/mol | 301.5 (193.25–523.5) | 141.0 (59.5–286.5) | **<0.001** | 250 (114–445) | 253 (133–394) | 0.708 |
| eGFR, ml/min/1.73m$^2$ | 40.5 (23.7–73.4) | 48.4 (27.2–83.4) | 0.137 | | | |
| IgA, g/L | 3.17 (2.49–4.19) | 4.09 (3.06–5.24) | **0.003** | 3.17 (2.36–4.15) | 3.84 (3.04–4.81) | **0.045** |
| C3, g/L | 1.25 (1.05–1.42) | 1.22 (1.01–1.42) | 0.679 | 1.22 (1.05–1.45) | 1.22 (0.97–1.44) | 0.575 |
| Haemoglobin, g/L | 120.5 (106.75–137.5) | 129 (113.5–142.0) | 0.056 | 121 (110–139) | 121 (105–140.5) | 0.911 |
| Albumin, g/L | 39.0 (35.25–42.0) | 39 (34.0–43.0) | 0.609 | 40 (35.5–42.5) | 38.5 (34.0–44.3) | 0.611 |
| ALP, U/L | 68.0 (59.5–79.25) | 71.0 (60.0–91.0) | 0.280 | 68 (60.0–79.0) | 81.0 (66.8–102.3) | **0.005** |
| P04, mmol/L | 1.21 (1.04–1.50) | 1.18 (1.01–1.34) | 0.316 | 1.10 (1.03–1.39) | 1.26 (1.09–1.42) | 0.134 |
| CCa, mmol/L | 2.18 (2.03–2.28) | 2.29 (2.20–2.35) | **<0.001** | 2.25 (2.10–2.33) | 2.22 (2.09–2.31) | 0.808 |
| Delta uPCR, mg/mmol/year | -16.8 (-46.87–11.07) | -2.65 (-14.56–5.50) | **0.003** | -12.7 (-35.5–12.4) | -7.12 (-26.7–1.50) | 0.904 |
| Delta eGFR, ml/min/1.73m$^2$ | -1.18 (-5.10–1.39) | -1.32 (-5.85–0.54) | 0.703 | -1.37 (-5.06–1.11) | -1.76 (-7.32–0.58) | 0.513 |
| ACEi/ ARB | 58 (84.1) | 234 (84.8) | 0.881 | 47 (82.5) | 47 (82.5) | 1.00 |
| RRT | 20 (29.0) | 74 (26.7) | 0.067 | 18 (31.6) | 26 (45.6) | 0.223 |
| Mortality | 9 (13.0) | 48 (17.3) | 0.391 | 7 (12.3) | 13 (22.8) | 0.140 |
| Follow up duration, months | 64.0 (26.0–97.5) | 60.0 (29.0–105.5) | 0.410 | 64 (28.0–98.0) | 39.0 (13.5–86.0) | 0.068 |

Continuous variables are presented as median (interquartile range), p-value by Mann–Whitney U-test. Categorical variables presented as number (percentage), p-value by Chi-square test.

In the matched cohort, propensity score matching utilising binary logistic regression analysis with a nearest neighbour approach was performed to match 57 patients who received immunosuppression with 57 who did not. Patients were matched for baseline hypertension, creatinine and proteinuria.

ACEi, angiotensin converting enzyme inhibitor; ALP, alkaline phosphatase; ARB, angiotensin receptor blockade; C3, complement 3; CCa, corrected calcium; CVD, cardiovascular disease; DBP, diastolic blood pressure; eGFR, estimated glomerular filtration rate; IgA, immunoglobulin; P04, phosphate; RRT, renal replacement therapy; SBP, systolic blood pressure; uPCR, urine protein creatinine ratio.

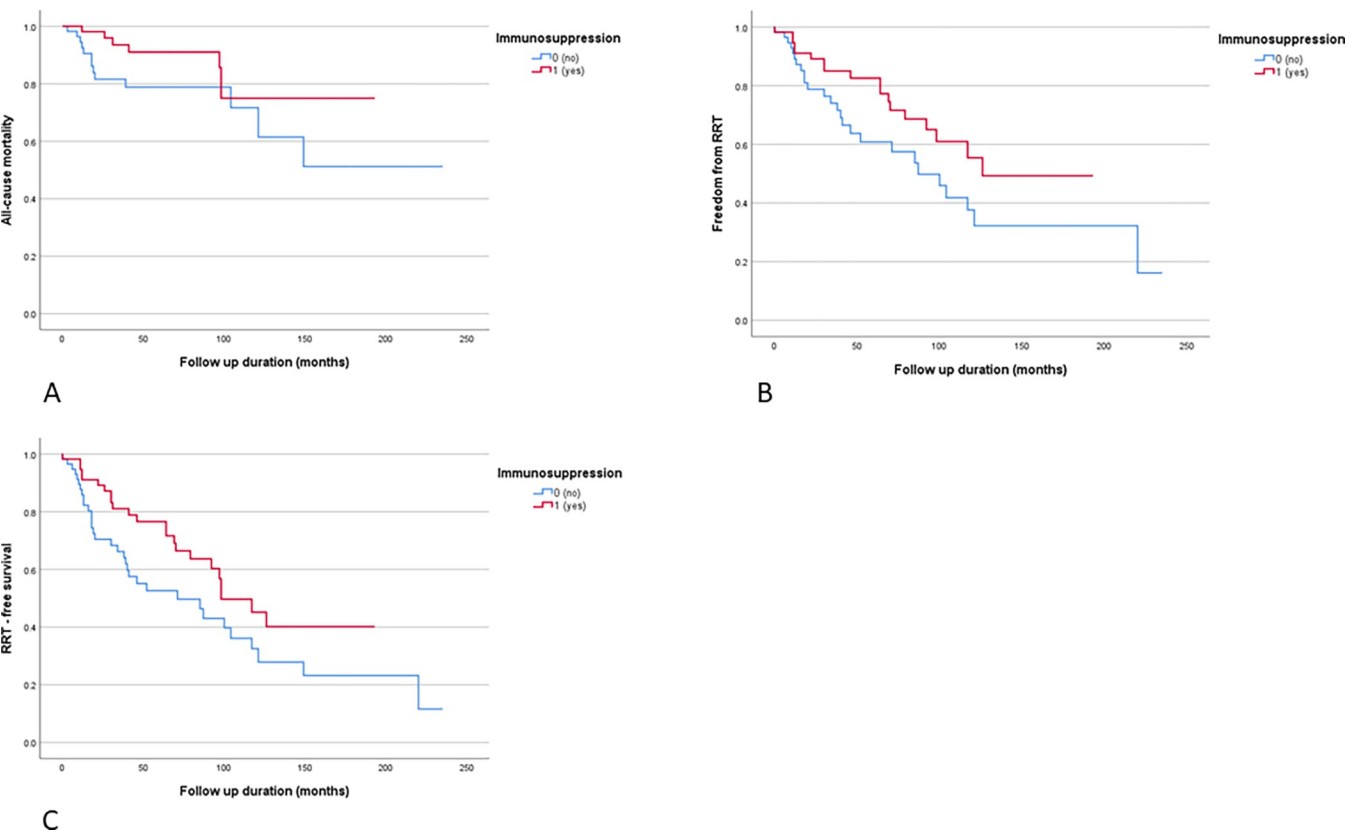

**Fig 3.** Kaplan-Meier curves for all-cause mortality (A), freedom from RRT (B) and RRT-free survival (C) for matched cohort comparing those who received immunosuppression and those who did not. P values 0.101, 0.074 and 0.051 respectively.

[13]. One laboratory test that we did not look at was uric acid, but it has been shown to predict poor outcomes in IgAN so this is something that may be worth assessing in future studies [14].

In our cohort, immunosuppression was used in a minority of patients (20.4%). There remains a significant risk of progression to ESKD over time (29.7% in our cohort). This is an underestimate of the true result given that some of our patients were diagnosed more recently and so will have a shorter duration of follow up.

We have demonstrated that those in the highest MEST-C score group have a higher rate of renal function decline, higher requirement for RRT and higher mortality. This correlates with previously published data [4] and demonstrates the value of using the MEST-C score when stratifying patients and making treatment decisions. We adopted the approach of analysis based on 'total MEST-C score', but it would also be interesting to split the cohort according to the presence of inflammatory (M, E and C) lesions or scarring (S and T) lesions.

Interestingly, in the univariate and multivariate models, immunosuppression was not associated with all-cause mortality or need for RRT. One hypothesis for this is that immunosuppression was generally used in those patients with more progressive disease, and that it ameliorated progression to such an extent that these patients had similar outcomes to those with milder disease.

We have shown that the average rate of eGFR loss in our cohort was -1.31ml/min/1.73m$^2$/ year. Whilst the more rapid decliners were more likely to require RRT, this did not translate into increased mortality. A previous study calculated eGFR slopes for the first-year post diagnosis of IgAN, and suggested that rapid and slow decliners over this period had significantly

increased risk of progression compared to non-decliners (relative risk 8.8 and 10.2 respectively) [15].

In the unmatched analysis of those who were given immunosuppression, the finding of higher rates of increased E, C and total MEST-C score in the immunosuppressed group indicates that the histological classification was taken into consideration. It is unclear why those who were given immunosuppression had a lower serum IgA level than those were not. Whilst it has been shown that measurement of galactose-deficient IgA1 (Gd-IgA1) and Gd-IgA1-containing immune complexes can aid with diagnosis and correlate with disease activity [16], use of the serum IgA concentration itself is not part of routine diagnosis or disease monitoring.

Whilst there was no statistical difference between immunosuppression use and outcomes in either the unmatched or matched group, there was a trend toward improved outcomes in the propensity matched cohort. It is important to note that the follow up duration was longer in the propensity matched immunosuppression group (64 vs 39 months), increasing the period over which outcome events can be recorded and thus potentially influencing results.

Immunosuppression reduced proteinuria levels more readily, but this did not translate into a difference in renal function decline. This correlates with the findings of STOP-IgAN which showed that immunosuppression improved rates of clinical remission by reducing proteinuria (full clinical remission achieved in 17% of cohort given immunosuppression, compared to 4% given supportive care, p = 0.01), but had no effect on overall renal function decline [10]. However, this contrasts with the findings from TESTING which showed both reduced proteinuria and slower renal function decline in the group given methylprednisolone (although the trial was stopped early due to increased incidence of adverse events in the immunosuppression group, and so conclusions about outcomes are more difficult to interpret) [9]. Recently presented data (not yet published) from the ongoing TESTING trial suggests that low dose methylprednisolone significantly improved primary outcomes (reduced major kidney outcomes by 47%) with a number needed to treat of just 6 patients to obtain benefit, with a significantly reduced incidence of serious adverse events (2.4 per 100 people treated). Publication of this data is eagerly awaited.

## Limitations

This was a retrospective observational study with the limitations of such a study design. Patients are likely to have started immunosuppression at different stages in their clinical journey, which may have had an impact on outcomes.

The MEST histological score was not introduced until 2012, with the addition of the C score several years later. As such, all MEST-C scores prior to 2012 had to be retrospectively determined. This was undertaken by an author (JS) using the biopsy report after having been trained in interpreting the report. Nevertheless, there may have been differences in reporting the MEST-C score between the renal pathologist and JS, although a proportion of MEST-C analyses were validated by an independent nephrologist.

## Conclusion

IgAN remains an important cause of ESKD. Treatment decisions require nuance given that there is no effective cure and the potential harm, as well as benefit, of utilising immunosuppression. Here, we show the benefit of taking into consideration histological scoring, as well as clinical characteristics, when making that decision. Whilst our study showed that immunosuppression did not improve overall requirement for RRT or mortality, there was a trend towards improved outcomes when duration of follow up was taken into consideration, suggesting that judicial use has a role.

## Supporting information

**S1 Table. Associations between baseline variables and RRT-free survival amongst those who were given immunosuppression (n = 82) using univariate and multivariate Cox regression analysis.**
(DOCX)

**S2 Table. Baseline characteristics, laboratory values and outcomes of cohort by rate of eGFR decline.**
(DOCX)

**S3 Table. Baseline characteristics and outcomes according to timing of biopsy- 2000–2011 vs 2012 onwards.**
(DOCX)

**S1 Fig.** Kaplan-Meier curves for all-cause mortality (A), freedom from RRT (B) and RRT-free survival (C) by rate of eGFR decline (>-5ml/min, -1 to -5ml/min and <-1ml/min). P-values 0.012, <0.001 and <0.001 respectively.
(TIF)

**S2 Fig.** Kaplan-Meier curves for 5 year (A) and 10 year (B) RRT-free survival by timing of biopsy (2000–2010 vs 2011–2019).
(TIF)

**S3 Fig.** Kaplan-Meier curves for 5 year (A) and 10 year (B) all-cause mortality by timing of biopsy (2000–2010 vs 2011–2019).
(TIF)

**S1 File. Effect of eGFR decline.**
(DOCX)

**S1 Dataset.**
(XLSX)

## Acknowledgments

We are very grateful to Ana Maria Aldanagelves and Schanhave Santhirasekaran (data scientists at Northern Care Alliance NHS Trust) for their assistance with some of the data analysis in this project.

## Author Contributions

**Conceptualization:** Smeeta Sinha, Philip A. Kalra.

**Data curation:** Joshua Storrar.

**Formal analysis:** Joshua Storrar, Rajkumar Chinnadurai.

**Investigation:** Joshua Storrar.

**Methodology:** Joshua Storrar.

**Supervision:** Smeeta Sinha, Philip A. Kalra.

**Writing – original draft:** Joshua Storrar.

**Writing – review & editing:** Smeeta Sinha, Philip A. Kalra.

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
