## [Decision Letter · Decision Letter 0]

21 Jun 2022

PONE-D-22-12533The epidemiology and evolution of IgA nephropathy over two decades: a single centre experiencePLOS ONE

Dear Dr. Storrar,

Thank you for submitting your manuscript to PLOS ONE. After careful consideration, we feel that it has merit but does not fully meet PLOS ONE’s publication criteria as it currently stands. Therefore, we invite you to submit a revised version of the manuscript that addresses the points raised during the review process.

We look forward to receiving your revised manuscript.

Kind regards,

Yasin Sahin

Academic Editor

PLOS ONE

Journal Requirements:

[I have read the journal's policy and the authors of this manuscript have the following competing interests.

S Sinha has received grants from Johnson and Johnson and AstraZeneca; speaker and lecture fees from AstraZeneca, Napp, Bayer, Sanofi-Genzyme, Novartis, and Vifor Pharma; is on advisory boards for Novartis, AstraZeneca, Bayer, and Travere; and has a clinical consultancy role with Sanifit. 

PA Kalra has received grants from Astellas, Vifor Pharma, BergenBio and Evotec; speaker and lecture fees from AstraZeneca, Napp, Bayer, Novartis, Vifor Pharma, Pharmacosmos, Boehringer Ingelheim; is on advisory boards for AstraZeneca and Vifor Pharma; and has a consultancy role with Bayer, Astella, Otsuka and Unicyte. 

JS and RC have no conflicts of interest to report.] 

5. Please include a copy of Table 4 which you refer to in your text on page 20.

Reviewers' comments:

Reviewer's Responses to Questions

**Comments to the Author**

1. Is the manuscript technically sound, and do the data support the conclusions?

Reviewer #1: Yes

Reviewer #2: Partly

Reviewer #3: Partly

2. Has the statistical analysis been performed appropriately and rigorously? 

Reviewer #1: Yes

Reviewer #2: No

Reviewer #3: Yes

3. Have the authors made all data underlying the findings in their manuscript fully available?

Reviewer #1: Yes

Reviewer #2: Yes

Reviewer #3: Yes

4. Is the manuscript presented in an intelligible fashion and written in standard English?

Reviewer #1: Yes

Reviewer #2: Yes

Reviewer #3: No

5. Review Comments to the Author

Reviewer #1: This retrospective study about IgA nephropathy in a single centre in UK is well-written and organized. It presents real data from every-day clinical practice over 20years. The authors have analyzed their data in a nice and clear way.

My main comments are

1.Fig 1 . 13 cases were excluded as"alternative diagnosis" . Does this mean non-IgA diagnosis or IgA plusa nother diagnosis?. Have the auhtors included in the final dialysis only IgAN ?Or some cases were IgA plus another GN, like diabetic or minimal change disease?

2.Were proteinuria measurements in 24hrurine collections or spots?In the second case, how many spots were available?

3.It is not clear how the decision for immunosuprresion was made. Does the centre use a specific protocole or KDIGO guidelines? Are the nephrologists free to decide on immunosuppresion or are they obliged to follow guidelines? Please specify

4. It would be interesting to analyze data before and after 2012, specially regarding treatment. Was any profound change in clinical practice during the two decades?

5. Moreover I would suggest analyze real data about ESKD vs International Risk-Prediction Tool in IgA Nephropathy(retrospectively).

Reviewer #2: The paper reports the longterm outcome in a UK cohort of patients with IgAN. I have a number of comments:

It should be clarified in the Introduction why the C score is not included in the risk prediction tool- (it added no value above including race i.e having crescents added the same risk as being East Asian)

I would be extremely cautious about analysing any data on MEST-C score that has been generated from reading a kidney biopsy report- I would remove this data from the analyses.

I wonder why you did not propensity match for age and gender?

Please define in the methods what medications you regarded/included as immunosuppression

There is no mention of a correction for multiple comparisons such as use of a Bonferroni correction- can you justify not correcting for multiple comparisons?

I am not sure what a "total MEST C score" means- there is no data in the literature on its biological/prognositic value or validity. It would have been much better to divide the lesions into chronic scarring lesions (S and T) and inflammatory lesions M/E/C if you wanted to look at clinicopathological associations.

Please describe the mortality in terms of deaths in the pre-and post development of ESKD- 1 in 5 of your IgAN patients died- this to me is very high- how does this relate to the studies from Scandinavia reporting survival stats in IgAN, and for comparison what are your local mortality rates for those with CKD? Life expectancy in Salford is amongst the lowest in the UK according to a recent report. According to the BHF the NorthWest has 2 local authorities with the highest rates of premature heart & circulatory disease death rates (2018-20). I think it is important to place your mortality stats in local context.

Reviewer #3: In the present retrospective, observational, single center study, the Authors described the clinical associates of kidney outcomes in 401 patients with biopsy proven IgA Nephropathy for a median follow-up of 51 months.

Patients were 45 years old with median baseline eGFR of 46.7 ml/min. The median decline of eGFR was -1.31ml/min/year and the median change in uPCR was -4.46mg/mmol/year. 29.7% progressed to RRT and 19.7% died.

1) The manuscript is poorly written and would be improved by a thorough English language review.

2) Data should be more extensively commented.

3) The available literature on the field has not been cited correctly, this makes the discussion not adequately constructive. Therefore, papers as following should be included in the Discussion section ( J Hypertens. 2020 May;38(5):925-935.AND Nutr Metab Cardiovasc Dis. 2020 Nov 27;30(12):2343-2350).

4) While the interpretation of the data seems extremely reasonable to the present reviewer, the retrospective study design is an insurmountable limitation. Nevertheless, limitations have been listed correctly and the sample size is one of the largest in the real world.

6. PLOS authors have the option to publish the peer review history of their article (what does this mean?). If published, this will include your full peer review and any attached files.

Reviewer #1: No

Reviewer #2: No

Reviewer #3: No

---

## [Author Response · Author response to Decision Letter 0]

26 Jul 2022

In response to your points:

1. The manuscript now meets PLOS ONE’s style requirements

2. We have added the statement “These do not alter our adherence to PLOS ONE policies on sharing data and materials” to the cover letter.

3. Minimal anonymized dataset has been uploaded as supporting information.

4. We have changed the affiliation of the corresponding author to The University of Manchester as well as the Northern Care Alliance NHS Foundation Trust.

5. The reference to table 4 on page 20 has been corrected to table 3.

Dear reviewers,

Many thanks for all of your comments. Our responses are detailed below.

Reviewer 1

1.Fig 1 . 13 cases were excluded as "alternative diagnosis". Does this mean non-IgA diagnosis or IgA plus another diagnosis?. Have the authors included in the final analysis only IgAN? Or some cases were IgA plus another GN, like diabetic or minimal change disease?

The ‘alternative diagnosis’ refers to a non-IgAN diagnosis that had been incorrectly labelled as IgAN (for example patients with increased serum IgA whose biopsy showed other pathology). Only exclusively IgAN cases have been included in the final analysis.

2.Were proteinuria measurements in 24hr urine collections or spots? In the second case, how many spots were available?

Proteinuria measurements were spot protein: creatinine ratios in all cases. Results were available for 337 patients at the point of diagnosis. 

3.It is not clear how the decision for immunosuppression was made. Does the centre use a specific protocol or KDIGO guidelines? Are the nephrologists free to decide on immunosuppression or are they obliged to follow guidelines? Please specify

This has changed over time. Prior to a dedicated glomerulonephritis clinic which was established in 2012 there was significant variation in what immunosuppression was used by individual clinicians who prescribed based upon their own experience and discretion. Since 2012, the clinic has been supported by a weekly multi-disciplinary meeting and treatment decisions have been more protocolled and where appropriate, evidence-based. For patients with progressive proteinuric IgAN we have tended to use prednisolone or prednisolone + MMF. 

4. It would be interesting to analyze data before and after 2012, specially regarding treatment. Was any profound change in clinical practice during the two decades?

We did look at this but found that there was no significant difference between the groups split 2000-2011 and 2012 onwards, other than that more immunosuppression was used in the latter decade- 25% of cases vs 16% of cases in the earlier decade. We have now included this as a supplementary table (S3 Table), which is referenced in the main manuscript. 

5. Moreover I would suggest analyze real data about ESKD vs International Risk-Prediction Tool in IgA Nephropathy (retrospectively).

Whilst this would be interesting, it was never our intention to validate the International IgA risk prediction tool. Rather, our aim was to describe the epidemiology and real world outcomes of a large cohort of patients with IgAN. Furthermore, many patients in our study were diagnosed in the last few years, and as such comparing their outcomes with risk of progression as per the International IgAN tool would not be possible at this point in time. 

Reviewer 2

It should be clarified in the Introduction why the C score is not included in the risk prediction tool- (it added no value above including race i.e having crescents added the same risk as being East Asian).

Thank you for pointing this out, we have added this to the introduction and included the reference.

I would be extremely cautious about analysing any data on MEST-C score that has been generated from reading a kidney biopsy report- I would remove this data from the analyses.

MEST-C scores have been routinely reported by our histopathologists since 2012. We understand that caution is needed with interpretation of kidney biopsy reports when generating the scores for biopsies pre-2012 but we would emphasise that the interpretation of data in the biopsy reports was cross-validated by two nephrologists for a proportion of cases, with a high degree of agreement. We believe that having MEST-C scores for the whole cohort of 401 patients has greater validity than only including those post 2012.

I wonder why you did not propensity match for age and gender?

The cohorts were matched for three major clinical parameters (baseline hypertension, creatinine, and proteinuria) based on a priori from previous observations. As the cohorts were already well matched for age and gender these were not included. We have now included this point in the methodology section to make this clear.

Please define in the methods what medications you regarded/included as immunosuppression

Many thanks. Immunosuppression treatment included prednisolone, cyclophosphamide, tacrolimus, ciclosporin, azathioprine and MMF (this has been added to the methods section).

There is no mention of a correction for multiple comparisons such as use of a Bonferroni correction- can you justify not correcting for multiple comparisons?

Throughout the analysis we have used non-parametric tests (Chi-square or Mann-Whitney U test) to identify the statistical difference in the characteristics between only two groups, hence we have not included the Bonferroni correction.

I am not sure what a "total MEST C score" means- there is no data in the literature on its biological/prognostic value or validity. It would have been much better to divide the lesions into chronic scarring lesions (S and T) and inflammatory lesions M/E/C if you wanted to look at clinicopathological associations.

Whilst a ‘total MEST C score’ has not been described in the literature, all of the individual features are taken into consideration in the International IgA risk prediction tool (with the exception of the C score). We used ‘total MEST C score’ as a way to divide the group based on histopathological features, but accept that a suitable alternative would have been to look at scarring and inflammatory lesions separately. We have commented on this in the discussion section.

Please describe the mortality in terms of deaths in the pre-and post development of ESKD- 1 in 5 of your IgAN patients died- this to me is very high- how does this relate to the studies from Scandinavia reporting survival stats in IgAN, and for comparison what are your local mortality rates for those with CKD? Life expectancy in Salford is amongst the lowest in the UK according to a recent report. According to the BHF the NorthWest has 2 local authorities with the highest rates of premature heart & circulatory disease death rates (2018-20). I think it is important to place your mortality stats in local context.

Whilst 1 in 5 patients with IgAN died, the high mortality rate can be explained by the follow up period of up to 20 years. As such some patients will have died from other causes, and not necessarily complications of CKD. 

To mitigate against the long follow up duration for some patients in this study (which may influence mortality rates) we have analyzed 5 and 10 year all-cause mortality as well as 5 and 10 year freedom from RRT for the cohort split according to decade of diagnosis. This can be seen in S2 Fig and S3 Fig. This demonstrates that there is no significant difference in mortality or progression to RRT depending on timing of entry into the study. Overall 5 year all-cause mortality was 10.2%, and 10 year all-cause mortality was 14.5%. This has been referenced in the results section of the manuscript.

Jarrick S et al. conducted a Swedish population based study assessing risk of mortality in over 3622 patients diagnosed with IgAN. They found that over a follow up period of 13.6 years, 577 patients died (15.9%) compared to 2066 (11.5%) in the reference population. This corresponded to a 6-year reduction in life expectancy for those with IgAN. They do report a 1.1% mortality rate in this study, but this is the rate of death per person-years, which is a different calculation. So, their rate of 15.9% is not particularly dissimilar to our rate of 19.7%, although they do have a longer average duration of follow up (13.6 years) compared to our cohort (4.3 years). However, the average age of their cohort was lower (median 38.8 years) compared to ours (45.0 years) and so you would expect resultant mortality to be lower. 

We have not been able to find any stats on local mortality rates for those with CKD. 

Reviewer 3

The manuscript is poorly written and would be improved by a thorough English language review.

We have reviewed the language again and believe that the manuscript is written in clear English. All of the authors were educated in the British educational system in the UK.

Data should be more extensively commented.

We have reviewed and extended our commentary regarding the data in the results section.

The available literature on the field has not been cited correctly, this makes the discussion not adequately constructive. Therefore, papers as following should be included in the Discussion section ( J Hypertens. 2020 May;38(5):925-935.AND Nutr Metab Cardiovasc Dis. 2020 Nov 27;30(12):2343-2350).

We believe that the literature has been cited correctly. However, these additional references have been reviewed and included. 

While the interpretation of the data seems extremely reasonable to the present reviewer, the retrospective study design is an insurmountable limitation. Nevertheless, limitations have been listed correctly and the sample size is one of the largest in the real world.

We agree that a retrospective study has limitations, however this is a very large real-world dataset which compares favourably in size and analytical methodology with other reports in the literature and we believe it adds value to clinicians managing patients with IgA nephropathy.

---

## [Decision Letter · Decision Letter 1]

5 Aug 2022

The epidemiology and evolution of IgA nephropathy over two decades: a single centre experience

PONE-D-22-12533R1

Dear Dr. Josh Storrar,

We’re pleased to inform you that your manuscript has been judged scientifically suitable for publication and will be formally accepted for publication once it meets all outstanding technical requirements.

Kind regards,

Yasin Sahin

Academic Editor

PLOS ONE

Additional Editor Comments (optional):

Reviewers' comments:

Reviewer's Responses to Questions

**Comments to the Author**

1. If the authors have adequately addressed your comments raised in a previous round of review and you feel that this manuscript is now acceptable for publication, you may indicate that here to bypass the “Comments to the Author” section, enter your conflict of interest statement in the “Confidential to Editor” section, and submit your "Accept" recommendation.

Reviewer #1: All comments have been addressed

Reviewer #2: All comments have been addressed

Reviewer #3: All comments have been addressed

2. Is the manuscript technically sound, and do the data support the conclusions?

Reviewer #1: Yes

Reviewer #2: Yes

Reviewer #3: Yes

3. Has the statistical analysis been performed appropriately and rigorously? 

Reviewer #1: Yes

Reviewer #2: Yes

Reviewer #3: Yes

4. Have the authors made all data underlying the findings in their manuscript fully available?

Reviewer #1: Yes

Reviewer #2: Yes

Reviewer #3: Yes

5. Is the manuscript presented in an intelligible fashion and written in standard English?

Reviewer #1: Yes

Reviewer #2: Yes

Reviewer #3: Yes

6. Review Comments to the Author

Reviewer #1: The authors have addressed my comments.

1.Fig 1 . 13 cases were excluded as "alternative diagnosis". Does this mean non-IgA diagnosis or IgA plus another diagnosis?. Have the authors included in the final analysis only IgAN? Or some cases were IgA plus another GN, like diabetic or minimal change disease?

The ‘alternative diagnosis’ refers to a non-IgAN diagnosis that had been incorrectly labelled as IgAN (for example patients with increased serum IgA whose biopsy showed other pathology). Only exclusively IgAN cases have been included in the final analysis.

2.Were proteinuria measurements in 24hr urine collections or spots? In the second case, how many spots were available?

Proteinuria measurements were spot protein: creatinine ratios in all cases. Results were available for 337 patients at the point of diagnosis.

3.It is not clear how the decision for immunosuppression was made. Does the centre use a specific protocol or KDIGO guidelines? Are the nephrologists free to decide on immunosuppression or are they obliged to follow guidelines? Please specify

This has changed over time. Prior to a dedicated glomerulonephritis clinic which was established in 2012 there was significant variation in what immunosuppression was used by individual clinicians who prescribed based upon their own experience and discretion. Since 2012, the clinic has been supported by a weekly multi-disciplinary meeting and treatment decisions have been more protocolled and where appropriate, evidence-based. For patients with progressive proteinuric IgAN we have tended to use prednisolone or prednisolone + MMF.

4. It would be interesting to analyze data before and after 2012, specially regarding treatment. Was any profound change in clinical practice during the two decades?

We did look at this but found that there was no significant difference between the groups split 2000-2011 and 2012 onwards, other than that more immunosuppression was used in the latter decade- 25% of cases vs 16% of cases in the earlier decade. We have now included this as a supplementary table (S3 Table), which is referenced in the main manuscript.

5. Moreover I would suggest analyze real data about ESKD vs International Risk-Prediction Tool in IgA Nephropathy (retrospectively).

Whilst this would be interesting, it was never our intention to validate the International IgA risk prediction tool. Rather, our aim was to describe the epidemiology and real world outcomes of a large cohort of patients with IgAN. Furthermore, many patients in our study were diagnosed in the last few years, and as such comparing their outcomes with risk of progression as per the International IgAN tool would not be possible at this point in time.

Reviewer #2: (No Response)

Reviewer #3: (No Response)

7. PLOS authors have the option to publish the peer review history of their article (what does this mean?). If published, this will include your full peer review and any attached files.

Reviewer #1: No

Reviewer #2: No

Reviewer #3: No

---

## [Editor Report · Acceptance letter]

22 Aug 2022

PONE-D-22-12533R1 

The epidemiology and evolution of IgA nephropathy over two decades: a single centre experience 

Dear Dr. Storrar:

I'm pleased to inform you that your manuscript has been deemed suitable for publication in PLOS ONE. Congratulations! Your manuscript is now with our production department. 

Kind regards, 

on behalf of

Dr. Yasin Sahin 

Academic Editor

PLOS ONE